# Targeting Breast Cancer Stem Cells Using Naturally Occurring Phytoestrogens

**DOI:** 10.3390/ijms23126813

**Published:** 2022-06-18

**Authors:** Mai Nguyen, Clodia Osipo

**Affiliations:** 1MD/PhD Program, Stritch School of Medicine, Loyola University Chicago, Maywood, IL 60153, USA; mnguyen16@luc.edu; 2Department of Microbiology and Immunology, Stritch School of Medicine, Loyola University Chicago, Maywood, IL 60153, USA; 3Department of Cancer Biology, Cardinal Bernardin Cancer Center, Stritch School of Medicine, Loyola University Chicago, Maywood, IL 60153, USA

**Keywords:** breast cancer, cancer stem cells, phytoestrogens, genistein, resveratrol

## Abstract

Breast cancer therapies have made significant strides in improving survival for patients over the past decades. However, recurrence and drug resistance continue to challenge long-term recurrence-free and overall survival rates. Mounting evidence supports the cancer stem cell model in which the existence of a small population of breast cancer stem cells (BCSCs) within the tumor enables these cells to evade conventional therapies and repopulate the tumor, giving rise to more aggressive, recurrent tumors. Thus, successful breast cancer therapy would need to target these BCSCs, as well the tumor bulk cells. Since the Women’s Health Initiative study reported an increased risk of breast cancer with the use of conventional hormone replacement therapy in postmenopausal women, many have turned their attention to phytoestrogens as a natural alternative. Phytoestrogens are plant compounds that share structural similarities with human estrogens and can bind to the estrogen receptors to alter the endocrine responses. Recent studies have found that phytoestrogens can also target BCSCs and have the potential to complement conventional therapy eradicating BCSCs. This review summarized the latest findings of different phytoestrogens and their effect on BCSCs, along with their mechanisms of action, including selective estrogen receptor binding and inhibition of molecular pathways used by BCSCs. The latest results of phytoestrogens in clinical trials are also discussed to further evaluate the use of phytoestrogen in the treatment and prevention of breast cancer.

## 1. Introduction

Breast cancer is the most common cancer in the United States, representing 14.8% of all new cancer cases in 2021 [1]. Due to early detection and improved treatments, the death rate for breast cancer has been falling continuously since 1992, and survival rate is very high for most patients [1]. However, challenges remain, as many patients may still relapse and die from long-term recurrence, metastasis and drug resistance [2,3]. Breast cancer is difficult to treat due to its inherent heterogeneity across the molecular, phenotypic and functional features within a patient’s tumor and across different patients [4]. Tumor heterogeneity across patients may be explained by intrinsic molecular subtypes of breast cancer, while inter-tumor heterogeneity can be explained with the cancer stem cell hypothesis [5]. In fact, increasing evidence points to the existence of a subpopulation of breast cancer stem cells (BCSCs) within a tumor. These cells can resist conventional therapies and repopulate the tumor, leading to relapse, recurrence and metastasis of more aggressive tumors [6]. Thus, to totally eradicate breast cancer and prevent relapse, strategies must be devised to address these BCSCs, as well as the bulk breast cancer cells. This review will briefly summarize the characteristics of BCSCs and then discuss the use of naturally occurring, dietary phytoestrogens as a potential BCSC therapy.

## 2. Characteristics of Breast Cancer Stem Cells

### 2.1. Mammary Stem Cells and Origin of BCSCs

Breast tissues contain a large pool of long-lived, self-renewing and multipotent mammary stem cells (MaSCs) that can give rise to both the luminal and basal epithelial lineages [4,7,8]. These MaSCs reside within a stem cell niche and are wired to respond to environmental cues, such as estrogen, Wnt, Hedgehog, Notch, TGF-β and growth factor signaling pathways [7,9,10]. Proper communication and response by MaSCs lead to the dynamic changes in the mammary glands seen during puberty, menstruation and pregnancy [7]. However, MaSCs are also at risk of acquiring mutations leading to transformation and tumorigenesis due to their inherent stem cell characteristics [9,11]. These MaSCs or progenitors give rise to a tumor subpopulation displaying dysregulated mammary stem cell characteristics called BCSCs [6,12,13,14,15]. Recent data support the existence of this small population of undifferentiated BCSCs with self-renewal and full differentiation capacity within the tumor [9,16]. In accordance with the cancer stem cell hypothesis observed in leukemia and in other types of cancer, this subpopulation of cancer stem cells is able to repopulate and reproduce the full heterogeneity of the original tumors [6,17].

### 2.2. Identification, Phenotypes and Roles of BCSCs in Breast Cancer

BCSCs, also known as tumor-initiating cells, comprise 0.1–1.0% of the tumor bulk according to various estimates across different breast cancer subtypes [5,15,18,19,20]. These BCSCs are identified as cancer cells expressing aldehyde dehydrogenase 1 (ALDH1^+^) and/or CD44^+^/CD24^−/low^ [21]. In 2003, Al-Haji et al. were the first to demonstrate that only a subpopulation of cells expressing the CD44^+^/CD24^−^/Lineage^−^ phenotype in human breast tumors was able to form tumors in immunodeficient mice [22]. Moreover, as few as 100 cells with this phenotype were capable of forming tumors in mice with similar heterogeneity to that of the original tumor, while tens of thousands of cells with other phenotypes failed to form tumors [22]. In vitro, these CD44^+^/CD24^−^ BCSCs displayed stemness properties, capable of self-renewal, extensive proliferation, clonal nonadherent spherical clusters (mammosphere formation), differentiation along mammary epithelial lineages and chemotherapy resistance [23,24]. Due to these stem-like characteristics, BCSCs were associated with poor prognosis and played critical roles in tumorigenesis, metastasis, recurrence and resistance to conventional therapy [5,15,19,25,26,27,28,29,30].

### 2.3. Signaling Pathways Critical for BCSCs

Studies have identified several key pathways, including Notch, Wnt/β-catenin, Hedgehog, Hippo, NF-κB, Stat3, HIF, TGF-β, PI3K/Akt, HER2 and BRCA1 as crucial in regulating the behavior and survival of BCSCs [2,9,31,32,33,34,35,36,37,38,39,40,41,42,43,44,45,46,47,48,49,50,51,52,53,54,55,56,57,58,59,60,61,62]. Specifically, embryonic developmental pathways, such as Notch, Wnt/β-catenin, Hedgehog and Hippo, normally regulate self-renewal and differentiation of normal mammary stem cells. Dysregulation or aberrant activation of these pathways transformed normal stem cells into BCSCs and led to tumorigenesis [9,60]. Inflammation-related pathways (NF-κB and Stat3), BCSC microenvironment and tumor hypoxia (TGF-β and HIF), proliferative pathways (HER2 and PI3K/Akt) and the loss of tumor suppressor (BRCA1) also contributed to the stem cell properties in BCSCs [9,60]. In addition to these pathways, recent studies showed that several microRNAs and long non-coding RNAs could modulate genes associated with these stem cell pathways to regulate BCSCs [2,9,15,63,64,65,66,67].

### 2.4. BCSCs in Resistance and Stem Cell Targeted Therapy

Current data identified BCSCs as key drivers responsible for resistance to first-line treatment, including conventional chemotherapy and radiotherapy, leading to treatment failure and cancer relapse [9,68]. BCSCs displayed inherent resistance to conventional therapy due to their high expression of drug transporters, plasticity and quiescence state, oxidative stress resistance, active DNA damage response and anti-apoptotic effects [2,28,29,69,70]. Thus, new targeted approaches are needed to eradicate BCSCs. One rational approach is to target the self-renewal pathways that BCSCs depend on, using Notch inhibitors (γ-secretase inhibitors), Wnt monoclonal antibodies, Hedgehog small molecule inhibitors, Hippo/mevalonate pathway inhibitors, HER-2 inhibitors and PI3K/Akt inhibitors [2,9]. Other strategies target the BCSC microenvironment via TGF-β inhibitors and CD44v6 monoclonal antibody, the DNA damage response, ABC drug transporters, miRNAs and the ubiquitin–proteasome system (UPS) [2]. CDK inhibitors, which could induce senescence phenotypes, or agents inducing stem cell differentiation are also being investigated [2,9]. Lastly, immunotherapy targeting PD-L1 and other novel BCSC antigens are being explored as another potential treatment venue [9]. A number of these therapies targeting stemness pathways are being tested in clinical trials with encouraging results, but none have yet been approved [9]. Toxicities and off-target effects on normal stem cells remain major concerns. Overall, these highlight the need for more effective and safer therapy against BCSCs.

### 2.5. Phytoestrogens and Natural Products Targeting BSCSs

Natural products, especially plant-derived products, have long been used in traditional medicine around the world [71]. Many well-known drugs used to treat breast cancer actually originated from plants, such as taxol, vincristine and vinblastine [72]. The ability to suppress BCSCs by inhibiting the stemness pathways was recently reported for several naturally occurring compounds, including salinomycin, curcumin, green tea EGCG, sulforaphane and piperine [73,74,75,76,77,78,79,80,81,82]. In this regard, one class of natural products targeting BCSCs rose to prominence: dietary phytoestrogens. These plant compounds with estrogenic activities have gained public attention since the publication of the Women’s Health Initiative in 2002. This study raised concerns about the use of estrogen and progestin in healthy postmenopausal women, as it was associated with significant adverse effects and increased risk of breast cancer [83]. Since then, hormone therapy usage sharply decreased, while herbal/dietary supplements usage increased [84]. As dietary supplements, phytoestrogens may be beneficial for menopause in older women or those unsuitable for hormone therapy [85]. Phytoestrogens have been shown to inhibit breast cancer growth and, more recently, they have been reported to inhibit BCSCs specifically. Hence, the rest of this review will dive deep into the use of several phytoestrogens as potential BCSC therapy.

## 3. Phytoestrogens Targeting BCSCs and Their Mechanisms of Action

Phytoestrogens were accidentally discovered in the 1940s due to a case of sheep grazing on red clover containing phytoestrogens, which resulted in their infertility [86,87]. These natural compounds are produced by various plants in response to environment stressors, serving as antioxidants, antifungals and antibiotics for the defense of plants [88,89]. Humans are also exposed to phytoestrogens via our regular diet. Incidentally, phytoestrogens are nonsteroidal polyphenols that share structural similarities with estradiol (17-β-estradiol) and bind both estrogen receptors with varying affinities, resulting in estrogenic and/or antiestrogenic effects [88]. Based on their chemical structures, phytoestrogens can be classified into four main classes: flavonoids, stilbenes, lignans and isoflavonoids [90]. Many studies have examined the use of various phytoestrogens in relieving menopausal symptoms, decreasing risks of hormonal cancers, osteoporosis and cardiovascular diseases [91]. This review solely focused on phytoestrogens that affect BCSCs, including the flavonoids (genistein, S-equol and naringenin) and the stilbenes (resveratrol and pterostilbene). A summary of these phytoestrogens and their mechanisms targeting BCSCs is depicted in Figure 1.

### 3.1. Flavonoids: Genistein, S-Equol and Naringenin

#### 3.1.1. Structures and Sources

One major group of dietary phytoestrogens is flavonoids. Flavonoids are composed of two aromatic rings bearing at least one hydroxyl group [88]. An important subclass of naturally occurring flavonoids is isoflavones, which are found mainly in soy and soy products. Major soy isoflavones include genistein and daidzein [88]. Genistein can be produced from the flavanone naringenin in soy plants [92]. Daidzein, upon consumption, can be further metabolized by microbes in the human gut to form the isoflavone S-equol. S-equol shows greater affinity for estrogen receptors than its precursor daidzein and is considered the more bioactive form [93,94]. However, studies estimated that only about 30% of the people in Western countries and 60% in Asian countries are able to produce S-equol following soy consumption, which might explain the different effects of soy observed in different studied populations [88,95].

#### 3.1.2. Estrogen Receptor (ER) Affinity

Since genistein shares structural similarities with 17-β-estradiol, it is known to activate both ERα and ERβ through a classical mechanism [96]. Unlike endogenous estrogens, which bind both ERs equally, genistein and S-equol both prefer ERβ over ERα [94,96]. A later study further showed that genistein (along with daidzein and equol) was more selective at enhancing ERβ-regulated genes at multiple levels, from higher ERβ affinity to higher efficiency at coactivator recruitment and chromatin binding [97]. While ERα is associated with stimulating proliferation of breast cancer cells, ERβ activation opposes ERα actions [98,99]. Thus, genistein’s preference for ERβ suggests that it could have dose-dependent effects on breast cancer cells, based on the expression of ERα compared to ERβ [88,100,101].

#### 3.1.3. Epidemiology

Soy isoflavones have gathered lots of interest in recent decades as potential chemopreventives. In fact, epidemiological studies associated higher intake of soy isoflavones in Asian countries (up to 47 mg/day) compared to Western countries (0.1–1.2 mg/day) with lower incidence of breast cancer [88,102,103]. Some studies further suggested that early soy/genistein exposure during childhood and adolescence might be needed to reduce breast cancer risk later in life [104,105,106,107,108]. In women with breast cancer, multiple Asian cohort studies found that soy food intake post-diagnosis was associated with lower mortality and cancer recurrence [109,110]. These observations suggest that soy isoflavones could inhibit breast cancer growth and recurrence, necessitating further studies to investigate causation and mechanisms.

#### 3.1.4. Genistein and Growth of Breast Cancer Bulk Cells

Genistein at low doses (≤10 μmol/L) was found to be estrogenic and promoted the growth of hormone-dependent breast cancer cell lines and tumors in mouse models [111,112,113,114]. This pro-tumorigenic effect is likely due to genistein acting as a weak estrogen, activating the ERα pathway. At a higher dose, genistein had been shown to inhibit breast cancer by inducing apoptosis, promoting cell cycle arrest and inhibiting angiogenesis [111,115,116,117,118]. These anticancer and anti-angiogenic effects of genistein are attributed to estrogen-independent pathways, including caspase activation, inhibition of VEGF signaling, PTK tyrosine kinase and MAPK inhibition and epigenetic modifications.

#### 3.1.5. Genistein and Its Role in BCSCs

Multiple studies showed that genistein inhibited BCSCs through direct inhibition of the pathways involved in stem cell growth and differentiation as well as paracrine signaling from surrounding cells. Montales et al. was the first study to report that both sera of adult mice fed with genistein and genistein itself were able to inhibit the basal stem-like CD44^+^/CD24^−^/ESA^+^ and the luminal progenitor CD24^+^ subpopulations from MDA-MB-231 and MCF-7 cells [119]. Furthermore, this inhibition of breast cancer mammosphere formation is associated with AKT inhibition and upregulation of PTEN [119]. In vivo, genistein given by intraperitoneal injection to nude mice-bearing MCF-7 xenografts was able to reduce BCSCs in the tumor by downregulating the Hedgehog–Gli1 self-renewal pathway [120]. Aside from direct effects, BCSCs may also be affected by the surrounding cells within the stem cell niche through paracrine signaling. Genistein inhibited mammary adipogenesis in vitro and in vivo, through the activation of ERβ and inhibition of PPARγ and *FASN* expression. Subsequently, the conditioned medium from these genistein-treated adipose cells inhibited mammophere formation of ER+ breast cancer cells (MCF-7), suggesting a paracrine effect [121]. Genistein at physiological concentrations (40 nM–2 μM) in co-cultures also stimulated ER+ breast cancer cells (MCF-7) to release amphiregulin. Released amphiregulin then induced the neighboring ER-negative BCSCs to differentiate into epithelial-like cells, correlating with the activation of PI3K/Akt and MEK/ERK signaling pathways [122]. In contrast to previous studies, Lauricella et al. demonstrated that treatment of MCF-7 cells with 25 μM genistein or 17-β estradiol actually increased the number and sizes of the mammosphere formed. Treatment with genistein increased the expression of ERα36 in tertiary mammospheres, enhancing their proliferation [123].

#### 3.1.6. S-Equol and Growth of Breast Cancer Bulk Cells and BCSCs

S-equol, which is the bioactive form of daidzein, has preference for ERβ over ERα, similar to that of genistein [94]. In fact, Yuan et al. showed that S-equol specifically induced ERβ tyrosine phosphorylation and inhibited breast cancer growth in vitro and in vivo [124]. However, S-equol also has dose-dependent effects on breast cancer cells, with higher S-equol concentrations (50–350 μM) inhibiting cancer growth and invasion [125,126,127,128,129,130]. At lower physiological concentrations (~1 μM), S-equol acted as weak estrogen to promote ER+ breast cancer proliferation in vitro but had no effect on tumor growth in mice [128,131,132,133]. In particular, the Dharmawardhane group showed that dietary daidzein increased breast tumor growth and metastasis in an ER-negative breast tumor mouse xenograft model [134]. This pro-tumorigenic effect can be recapitulated in vitro by the daidzein metabolite, S-equol (25–50 μM), increasing breast cancer proliferation of ER-negative breast cancer cells via upregulation of c-Myc, eIF4GI and enhanced protein synthesis [135]. A subsequent study showed that S-equol also increased the size and number of mammospheres in ER-negative MDA-MB-435 cells through the upregulation of c-Myc [136]. This suggests that S-equol in certain situations could enhance BCSCs in ER-negative breast cancer. However, data on S-equol in BCSCs are severely lacking and would merit further study to delineate the mechanism of action by S-equol.

#### 3.1.7. Naringenin and Growth of Breast Cancer Bulk Cells and BCSCs

Naringenin is the precursor of genistein [92]. Naringenin (4′,5,7-trihydroxy flavanone) and naringin (4,5,7-trihydroxy flavonone 7-rhamnoglucoside) are the major flavanones found in citrus fruits, such as grapefruits, oranges, cherries and tomatoes [137,138,139]. In humans, the glycoside naringin is metabolized by gut bacteria into the aglycone naringenin [137,139,140,141]. Naringenin is considered the more bioactive form due to the lack of steric hindrance of the sugar moiety [142]. Similar to other phytoestrogens, naringenin weakly binds both ERα and ERβ, with a preference for ERβ (relative binding ERβ/ERα = 0.5%/0.08%, E2 = 100%) [143]. Naringenin was found to inhibit the growth and migration of both ER-negative (MDA-MB-453, MDA-MB-231) and ER-positive (MCF-7) breast cancer cells in vitro and in vivo [142,144,145,146,147,148,149,150,151,152,153]. Furthermore, combining tamoxifen with naringenin more effectively inhibited the proliferation of ER-positive breast cancer cells via inhibition of both PI3K and MAPK pathways and modulation of several ER-target genes [152,154]. Naringenin is also effective at targeting BCSCs, as reported in recent studies. Naringenin inhibited BCSCs both in vitro and in vivo in a DAXX-dependent manner [155]. Naringenin’s upregulation of DAXX and subsequent inhibition of BCSCs were more selective toward ERβ than ERα [155]. Similarly, Hermawan et al. found that naringenin inhibited BCSC mammosphere formation, potentially through the modulation of p53 and ERα mRNA [82]. Naringenin was also identified as the active compound in the Xihuang pill, a traditional medicine used to treat breast cancer in China [156]. In vitro, both the Xihuang pill and naringenin inhibited mammosphere formation in triple negative breast cancer cell lines by upregulating NR3C2 expression [156].

### 3.2. Stilbenes: Resveratrol and Pterostilbene

#### 3.2.1. Structures and Sources

Stilbene is a class of phytoestrogens bearing two benzene rings joined by an ethanol or ethylene [157]. Arguably, the most well-known stilbene is resveratrol, which is a metabolized stilbene naturally produced by several plants in response to pathogen attacks [158,159]. *Trans*-resveratrol is the more dominant, estrogenic form due to its configuration, and it can be found in peanuts, blueberries, pines, grapevine and especially red wine [90,160,161,162]. Pterostilbene is a natural methoxylated derivative of resveratrol found in similar plants [163,164]. Pterostilbene is considered the more bioavailable and bioactive molecule compared to resveratrol [165,166].

#### 3.2.2. Estrogen Receptor (ER) Affinity

*Trans*-resveratrol shares structural similarity with diethylstilbestrol (a synthetic human estrogen) and binds both ERα and ERβ equally, unlike other phytoestrogens, which preferentially bind ERβ [167]. However, resveratrol binds ERs with 7000-fold lower affinity than estradiol and can act as a weak estrogen in the absence of 17β-estradiol [167,168]. Resveratrol competes with 17β-estradiol for ERα on specific EREs but not ERβ [167]. These results indicate that resveratrol could differentially alter the activity of ERα and ERβ as an agonist or antagonist [167,169]. Although no direct ER binding data on pterostilbene were found, recent studies reported that the anticancer and antioxidant effects of pterostilbene required ERα and/or ERβ [170,171,172,173]. This suggests that pterostilbene could also alter ER activity due its structural similarity with resveratrol.

#### 3.2.3. Epidemiology

The Mediterranean diet, which includes significant plant polyphenols and wine consumption, has been associated with lower incidence of breast cancer [174,175,176,177,178]. Resveratrol has been identified as one of the protective bioactive molecules in this diet [179]. Additionally, resveratrol is thought to be the ingredient in wine responsible for the “French Paradox” effect, in which moderate wine consumption in French people is associated with low incidence of heart disease despite their high-fat diet [90,160,180,181]. When resveratrol concentration was examined, studies reported some conflicting results. In the Swiss Canton of Vaud case study, Levi et al. found a significant inverse association between resveratrol intake from grape consumption and breast cancer risk [182]. Meanwhile, the Chianti cohort study found no association between urinary resveratrol and cancer-related mortality in older adults [183]. Overall, resveratrol still gathers significant interest as a chemopreventive agent with many benefits.

#### 3.2.4. Resveratrol and Growth of Bulk Breast Cancer Cells

Resveratrol exhibits a dose-dependent, biphasic effect on ER+ breast cancer cells in vitro [184]. Various studies reported that resveratrol stimulated proliferation of ERα+ breast cancer cells at low doses (<10–22 μM) but inhibited proliferation and induced cell death at higher doses (>10–22 μM) [185,186,187,188,189]. For ER-negative breast cancer cells, resveratrol inhibited proliferation at both low and high doses [189,190,191,192,193]. Resveratrol inhibited breast cancer cell migration and invasion induced by nearby cancer-associated fibroblasts [194]. In vivo data from mouse and rat models showed that resveratrol treatment decreased breast tumor initiation and onset, tumor growth and angiogenesis, likely through the inhibition of oxidative DNA damage, tumor-promoting enzymes and/or NF-κB signaling [168,195,196,197,198,199].

#### 3.2.5. Resveratrol and BCSCs

Recent studies found that resveratrol targeted BCSCs via multiple signaling pathways [165]. Fu et al. showed that resveratrol significantly inhibited survival of BCSCs and reduced the number and size of mammosphere formed in vitro mediated by induction of autophagy and suppression of Wnt/β-catenin signaling [200]. Injection of resveratrol in vivo likewise inhibited xenograft tumor growth and reduced the BCSC population [200]. In another study, resveratrol reduced BCSCs growth as assessed by mammosphere formation and inhibited tumor growth in xenograft mice models by suppressing lipogenesis and inducing Fas-mediated apoptosis [201]. Singh et al. reported that resveratrol inhibited mammosphere formation and breast carcinogenesis in the presence of estradiol in rats by inducing the NRF2 pathways [202]. A recent study by Peiffer et al. identified that resveratrol inhibited survival of BCSCs from ER+ breast cancer cells in a DAXX-dependent manner [155]. Additionally, resveratrol suppressed self-renewal activity of BCSCs mediated by cancer-associated fibroblasts through inhibition of Bmi-1 and Sox2 expression [194]. Alternatively, resveratrol has been shown to promote Argonaute2 (a central RNAi component) activity and enhanced tumor-suppressive miRNAs to inhibit BCSCs [203].

#### 3.2.6. Pterostilbene and Growth of Bulk Cells and BCSCs

As a natural analog of resveratrol, pterostilbene has antioxidative activity by inhibiting COX-1 and has been shown to inhibit carcinogen-induced preneoplastic lesions in mammary organ cultures [204]. Although data are lacking, pterostilbene is thought to be similar to resveratrol by inhibiting cancer cell proliferation and inducing apoptosis [205]. In HER2+ breast cancer cells, pterostilbene inhibited HER2-mediated invasion and metastasis through downregulation of MMP-9 expression and inhibition of p38 and Akt pathways [205]. Pterostilbene has been shown to inhibit BCSCs via Argonaute2-dependent mechanism in a similar manner as resveratrol [203]. Additionally, pterostilbene also suppressed the generation of BCSCs and metastatic potential induced by tumor-associated macrophages [206]. This suppression is potentially mediated by the modulation of the NF-κB/miR488 circuit [206]. Wu et al. found that pterostilbene selectively killed BCSCs isolated from MCF-7, inhibited mammosphere formation and enhanced BCSCs sensitivity to chemotherapeutic drugs through reduction in CD44 expression and inhibition of the hedgehog/Akt/GSK3β signaling [207].

## 4. Clinical Trials Investigating Effects of Phytoestrogens on Breast Cancer

Based on the promising results of phytoestrogens inhibiting breast cancer in vitro and in vivo, several clinical trials investigated the protective effects of phytoestrogens in breast cancer treatment and prevention. Table 1 comprehensively listed the results of trials on the NIH website that was published between 2008 and 2020. Most phytoestrogens tested were soy isoflavones and its metabolite, S-equol, or flaxseed lignan and its metabolite, enterolactone. As expected, soy isoflavone is the most studied phytoestrogen and the most tested in clinical trials. Soy earned its fame from an epidemiological finding in which women living in Asian countries have lower incidence of breast cancer compared to Western women, and this is thought to be attributed to their diet containing soy phytoestrogens [103,208]. As soy is known to have estrogenic properties, many studies since have investigated the effects of soy isoflavones on breast tissue and breast cancer. However, the results from clinical studies have been conflicting, with soy isoflavones inhibiting breast epithelial cell growth in some trials, stimulating proliferation or having little effect in others. A recent study, including 39 patients with invasive triple negative breast cancer, demonstrated that a short course of oral S-equol was sufficient to inhibit the proliferation of breast tumor cells, as measured by Ki-67, with nearly one-third of patients having up to 20% decrease in Ki-67 expression [209]. Likewise, a prospective study on a cohort of German postmenopausal patients with breast cancer found that higher serum genistein was associated with lower Ki-67 expression in tumors [210]. Some studies further suggested that soy isoflavone supplement in healthy women could reduce breast cancer risk by inducing favorable modification of estrogen metabolism, lowering bone density and modulating blood pressure [211,212,213]. In contrast, Khan et al. found that an oral form of soy isoflavone containing genistein and daidzein did not reduce breast epithelial proliferation in healthy Western women who were at high risk of developing breast cancer; instead it induced a 27% increase in Ki-67 in premenopausal women [214]. Similarly, Shike et al. found soy intake in women with invasive adenocarcinoma induced genes that drive cell cycle progression and proliferation pathways in breast tumor cells, although no significant change was observed in Ki-67 expression [215]. Aside from proliferation, multiple studies found that soy isoflavones were well tolerated even at high dosages and did not induce adverse changes in breast density, DNA damage or apoptosis [216,217,218]. The discrepancy in these studies could be due to dosages of different soy isoflavone mixtures, duration of treatment, individual abilities to metabolize isoflavones, the composition of isoflavone-metabolizing microbiome, menopausal status and/or incidence of breast cancer. For example, people who can produce equol from soy isoflavones can exhibit different responses with soy consumption [219]. Recently, a small study in healthy Belgian and Dutch women found that oral consumption of soy products containing isoflavones led to high levels of genistein and daidzein, specifically in the breast tissue [220], suggesting they may be sufficient to have direct effects on breast tissue. More studies would be needed to delineate which patient population would benefit from soy isoflavone supplements and when.

Aside from soy isoflavones, red clover isoflavones were found to have no significant effects on breast density, endometrial thickness, serum cholesterol, follicle stimulating hormone levels and bone mineral density in a randomized trial of 401 healthy women in the United Kingdom [221]. Similarly, in a double-blinded, randomized trial of 152 premenopausal women at high risk of breast cancer, flaxseed Secoisolariciresinol Diglycoside (SDG) was reported to have no effect on Ki-67 expression in breast tissues [222]. This finding is different from the previous open-labeled, pilot study from the same group in which 49 healthy women took SDG supplement for a year and found their Ki-67 expression reduced from 4% to 2% [223]. Flaxseed had no effect on serum hormone levels or prognostic breast tumor characteristics when combined with an aromatase inhibitor in postmenopausal women with ER+ breast cancer [224].

**Table 1 ijms-23-06813-t001:** Reported results of clinical trials on phytoestrogens and breast cancer (2008–2020).

Phytoestrogen	Trial Design	Sample Size/	Interventions	Results	NCT Number/
		Population Studied			References
Enterolactone and Genistein	Observational prospective cohort study	MARIE cohort of 1060–2105 postmenopausal breast cancer patients in Germany Aged 50–74 years	Lifestyle questionnaires and blood samples collected at recruitment 2002–2005 (baseline) and 2009 (follow-up)	Higher genistein concentrations were associated with lower Ki-67 expression in tumors showing >20% Ki-67 No associations between enterolactone or genistein and HER2 status Enterolactone concentration inversely associated with all- cause mortality, breast-cancer-specific mortality and distant disease-free survival, likely through mediation of C-reactive protein Higher enterolactone concentrations were associated with improved 5-year survival for postmenopausal breast cancer patients up to 4 years post-diagnosis Higher concentrations of genistein, resveratrol and luteolin at follow-up in long- term survivors were associated with poorer subsequent prognosis	NCT03401034 (Jaskulski et al., 2017 [209]) (Jaskulski et al., 2018 [225]) (Jaskulski et al., 2020 [226])
Estrogenic Botanical Supplements	Observational study	Up to 3159 women in the UK with invasive primary breast cancer at 9–15 months post-diagnosis Aged 18 to 75 years	Questionnaires (diet, lifestyle, use of complementary treatments) and blood/urine samples were collected annually for up to 5 years	Estrogenic botanical supplement usage doubled after diagnosis (8.4%) Flaxseed and soy/isoflavone were most commonly used Pre-diagnosis phytoestrogen intake was not associated with factors associated with improved breast cancer prognosis	NCT00701584 (Velentzis et al., 2011 [224]) (Swann et al., 2013 [227])
Red Clover Isoflavones	Double-blind, randomized intervention study	401 healthy women in the UK with at least one first-degree relative with breast cancer, in the UK Aged 35 to 70 years	Red clover isoflavones for 3 years	Red clover isoflavones were well tolerated in healthy women No significant differences in breast density, endometrial thickness, serum cholesterol, follicle stimulating hormone levels and bone mineral density	(Powles et al., 2008 [220])
Flaxseed Lignan Secoisolaricires inol Diglycoside (SDG)	Double-blind, randomized intervention study, phase IIB	152 premenopausal women who have a >2-fold relative risk of breast cancer compared to women in their age group, in USA Aged 21–49 years	50 mg of (SDG) capsule once daily for 12 months	No difference in breast epithelial cells’ Ki-67 expression between SDG and placebo	NCT01276704 (Fabian et al., 2020 [221])
Flaxseed (with aromatase inhibitor)	2 × 2 factorial, randomized interventional study	24 postmenopausal women with estrogen receptor positive (ER+) breast cancer receiving surgery at Roswell	25 g/day ground flaxseed +/−1 mg/day anastrozole for 13–16 days prior to breast surgery	No interaction between flaxseed and aromatase inhibitor anastrozole in serum hormone levels or prognostic breast tumor characteristics	NCT00612560 (McCann et al., 2014 [223])
		Park Cancer Institute, USA Aged 59–65 years		Anastrozole may reduce circulating lignans induced by flaxseed	
S-equol	Open-label intervention study, early phase I	39 patients in Texas, USA, with invasive triple-negative breast cancer, confirmed by core needle biopsy Aged 18 and older	S-equol at a dose of 50 mg or 150 mG PO twice daily for 10–21 days	S-equol was well tolerated and inhibited proliferation of breast tumor cells, as measured by a decrease in Ki-67 (8% compared to baseline) Up to 20% decrease in Ki- 67 was observed in 28% of S-equol-treated patients	NCT02352025 (Lathrop et al., 2020 [208])
Soy Isoflavones	Randomized intervention study	31 healthy Belgian or Dutch women who were scheduled for an esthetic breast reduction Aged 18 to 62 years	Soy milk (16.98 mg genistein and 5.40 mg daidzein aglycone equivalents per dose) or soy supplement (5.27 mg genistein and 17.56 mg daidzein aglycone equivalents per dose), with three doses daily for 5 days before breast reduction	After soy product intake, genistein and total daidzein concentrations reached high levels in breast tissue, which could be sufficient to cause potential health effects	(Bolca et al., 2010 [219])
Soy Isoflavones	Double-blind, randomized intervention study	85 previously treated breast cancer women at high risk of breast cancer living in CA, USA Aged 30–75 years	Oral soy isoflavones (50 mg/day) for 12 months	Treatment increased plasma soy isoflavone levels with minimal adverse effect Soy supplementation did not decrease mammographic density	NCT01219075 (Wu et al., 2015 [215])
Soy Isoflavones	Double-blind, randomized	80 postmenopausal women with	250 mg of standardized soy extract corresponding to	Soy isoflavones did not affect breast density as measured by	(Delmanto et al., 2013 [216])
	intervention study	vasomotor symptoms in Brazil Aged >45 years	100 mg/day isoflavone for 10 months	mammography and ultrasound	
Soy Isoflavones	Randomized, placebo-controlled intervention study	140 women with invasive breast adenocarcinoma in NY, USA Aged mean 56 ± 12 years	5.8 g soy protein powder twice a day for 7–30 days prior to breast surgery	Soy intake induced overexpression of FGFR2 and genes that drive cell cycle and proliferation pathways in breast tumor cells No significant changes in Ki67 or Caspase3	(Shike et al., 2014 [214])
Soy Isoflavones	Double-blind, randomized intervention study	200 healthy premenopausal women in TX, USA Aged 30 to 42 years	Soy isoflavone tablet (60 mg daidzein, 60 mg genistein and 16.6 mg glycitein) twice per day for five days per week for up to 2 years	Isoflavones tended to normalize systolic blood pressure via serum calcium moderation and decreased diastolic blood pressure, independent of calcium level Genistein significantly decreased whole-body bone mineral density at low serum calcium levels	NCT00204490 (Lu et al., 2020 [212]) (Nayeem et al., 2019 [211])
Soy Isoflavones/Genistein	Double-blind, randomized intervention study, phase IIB	126 healthy women who were at increased risk of developing breast cancer in IL, USA Aged 42–55 years	Oral PTI G-2535 pill (genistein 150 mg, daidzein 74 mg, glycitein 11 mg) once daily up to 6 months	Soy isoflavones in healthy, high-risk adult Western women did not reduce breast epithelial proliferation, as measured by Ki-67 In premenopausal women, soy induced a 27% increase in Ki-67 in breast epithelial cells post-intervention	NCT00290758 (Khan et al., 2012 [213])
Soy Isoflavones/Genistein/Soy Isoflavones/Genistein	Double-blind, randomized intervention study, phase	30 healthy non- obese postmenopausal women at no risk of breast cancer, living in NC, USA Aged 45–70 years	Oral genistein (PTI G-2535) twice daily for 84 days	High dose of soy isoflavones (900 mg) in postmenopausal women did not cause DNA damages, apoptosis or significant estrogenic effects	NCT00099008 (Pop et al., 2008 [217])
Soy Protein (with seaweed)	Double-blind, randomized with crossover intervention study	15 healthy postmenopausal European–American women living in central MA, USA Aged mean 58.8 ± 7.9 years	7 weeks of 5 g/day seaweed (Alaria), plus 2 mg isoflavones/kg body weight during week 7; crossover after 3-week washout	Soy and SeaSoy (seaweed plus soy) significantly decreased serum E1, increased urinary excretion of estrogen metabolites and altered phytoestrogen metabolism	NCT01204957 (Teas et al., 2009 [210])
Genistein (with Gemcitabine)	Open-label intervention study, phase II	17 women with metastatic stage IV breast cancer in MI, USA Aged 31 to 57	Oral genistein (100 mg) once daily on days −7 to 1, then twice daily from days 1 to 21; IV gemcitabine hydrochloride (1000 mg/m^2^) on days 1 and 8; course repeated up to 24 weeks	Study was closed early due to lack of efficacy	NCT00244933

An important measure would be to assess whether phytoestrogens regulate survival of BCSCs. For example, either recurrence-free survival or overall survival rates would be appropriate endpoints. However, there are few studies that have reported recurrence-free or overall survival outcome results, as time to these outcomes requires years of observations, and these phytoestrogen studies are rather recent. An observational study on a large cohort of ~3159 British women with invasive primary breast cancer reported that women changed their diet after diagnosis and doubled their usage of estrogenic botanical supplement (8.4% of the cohort), with flaxseed and soy isoflavone being the most common [224]. However, pre-diagnosis phytoestrogen intake was not associated with improved breast cancer diagnosis [227]. In a cohort of German postmenopausal breast cancer patients, higher serum enterolactone concentration was associated with lower all-cause mortality, breast-cancer-specific mortality and higher distant disease-free survival [225]. This increase in survival is likely mediated through enterolactone’s modification of the inflammatory marker C-reactive protein [225]. Higher concentrations of enterolactone were also associated with improved 5-year survival for patients in this cohort; however the association was only up to 4 years post-diagnosis [226]. On the other hand, genistein, resveratrol or luteolin concentrations in survivors after long-term follow-up were associated with poorer subsequent diagnosis [226]. So far, only enterolactone is reported to be associated with improved survival in breast cancer patients in observational studies. A number of studies on phytoestrogens, especially soy isoflavones, have yet to report their long-term survival outcomes. It will be important to examine the effect of interventional, placebo-controlled phytoestrogens on survival and long-term recurrence to see whether they would be effective at targeting BCSCs.

## 5. Conclusions

Although breast cancer treatment has significantly improved patient survival, the latest frontiers of breast cancer research focus on preventing metastasis and recurrence. As more studies unraveled the role and significance of BCSCs in resistance, recurrence and metastasis, it became evident that we must target BCSCs to completely eradicate breast cancer at the root. Numerous therapies tackling BCSCs, from stemness pathway inhibitors to immunotherapies, are being studied and tested in clinical trials. Nonetheless, there is much interest in using diet-based, natural products as chemopreventives and/or treatment adjuvants. Natural products are thought to be less toxic, cause fewer side effects, cheaper and more available/accessible to patients worldwide. Additionally, epidemiological studies report very encouraging results associating certain diets containing soy and other phytoestrogens with lower breast cancer incidence and better outcome. Among natural products studied, phytoestrogens (particularly genistein, S-equol, naringenin, resveratrol and pterostilbene) have emerged as key players with the ability to inhibit BCSCs through various mechanisms.

However, recommendations by primary physicians or oncologists regarding phytoestrogen consumption for pre/postmenopausal women or healthy/cancer patients remain unclear. There are questions about the risk to benefit ratio and which population would benefit from phytoestrogen supplementation. Some data are conflicting between different studies concerning the effect of phytoestrogens on breast cancer. Due to the biphasic nature of phytoestrogens and their binding to ERs, many phytoestrogens exhibited opposing effects on breast cancer in vitro and in vivo depending on the dosage. Furthermore, there are significant differences in phytoestrogen metabolism between mice and humans and between individuals, which further complicate the issue of dosing needed to achieve the desired clinical outcomes. The off-target effects of phytoestrogens on normal stem cells must also be considered, as these cells share many similarities with BCSCs. In addition, there is a lack of controlled interventional studies with phytoestrogens and long-term recurrence-free survival and overall survival outcomes. Although the data thus far are promising, more studies are needed to understand the mechanisms, the interactions with surrounding cells and other breast cancer treatments, as well as the effective clinical dosage in order to establish phytoestrogens as a viable anti-BCSC therapy. Ultimately, the hope is that patients could easily supplement their diet with phytoestrogens to prevent breast cancer and/or recurrence, with little toxicity and side effects.

## Figures and Tables

**Figure 1 ijms-23-06813-f001:**
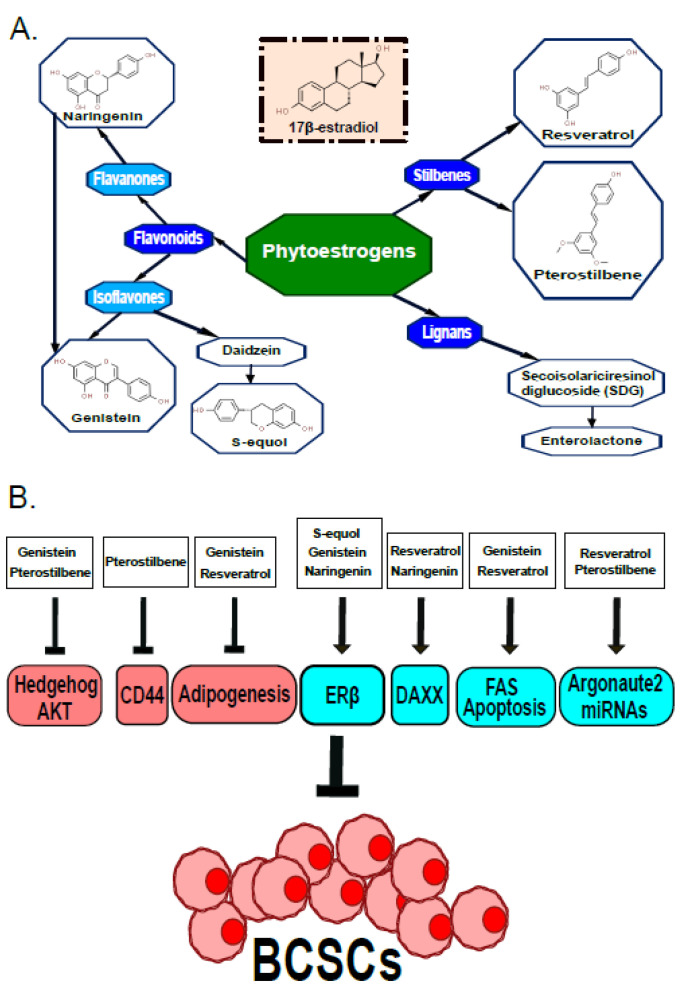
**Structure and Function of Phytoestrogens on Breast Cancer Stem Cells.** (**A**). Chemical structures of different classes of phytoestrogens compared to 17β-estradiol. Colors highlight different subclasses of phytoestrogens. (**B**). Phytoestrogens either inhibit or activate different signaling pathways to suppress breast cancer stem cells (BSCS). Specifically, inhibition of hedgehog, CD44 or adipogenesis by genistein, pterostilbene or resveratrol reduces survival of BCSC. Activation of ERβ, DAXX, FAS, argonaute2 or miRNAs by S-equol, genistein, naringenin, resveratrol or pterostilbene suppresses survival of BCSC. Red indicates inhibition of pathways. Blue indicates activation of pathways.

## Data Availability

Not Applicable.

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
