# Peer review of "Targeting Breast Cancer Stem Cells Using Naturally Occurring Phytoestrogens"

_ijms, 2022, doi:10.3390/ijms23126813_

Round 1

Reviewer 1 Report

Breast cancer is heterogeneous and is divided into several subtypes, the authors is encouraged to specifically describe in more detail how these phytoestrogens affect BCSCs and particular subtype through various relevant signaling pathways and the underlying mechanisms, such as the effects on lineage conversion between basal, luminal, BCSCs, mammary stem cells, and the behind cancer driver genes or stemness and fate-determinant factors other than Ki-67 (PMID: 26340526, PMID: 28062556, PMID: 31013830). The effects of phytoestrogens on advanced cancer with heterogeneity are also the concerned issues.

More importantly, the authors should justify the strength of evidence provided by the literatures instead of just mention the observed results of these studies, even though the authors list more than 200 references. Actually, many results of these studies are controversial because these results are influenced by many confounding factors that should be included for consideration and discussed.

It is suggested to provide authors’ personal opinions for the future direction of investigation to overcome such controversial effect of phytoestrogen on BCSCs.

3.1.4 “These anticancer effects of genistein are attributed to estrogen-independent pathways, such as caspase activation, tyrosine kinase inhibition, and epigenetic modifications.” à explain more, what kinds of epigenetic modifications? What tyrosine kinases are inhibited?

Have a table for in vitro studies to show the differential dose-effects of these phytoestrogens because many of them exhibit both tumor promotion and suppression effects depending on the doses. Please also indicate the concentrations and the cell types when describe their biological activities, such as in section 3.1.7, “Naringenin was found to inhibit the growth and migration of both ER-negative and ER-positive breast cancer cells in vitro and in vivo…”

Figure 1B legend, can be explained in more detail.

Tables are too small to read.

Typo error:

Page 6, section 3.1.5, the neighboring ER-“nehative” BCSCs

Author Response

Dear Reviewer,

Thank you for your valuable comments and suggestions. In answer to your specific comments:

Breast cancer is heterogeneous and is divided into several subtypes, the authors is encouraged to specifically describe in more detail how these phytoestrogens affect BCSCs and particular subtype through various relevant signaling pathways and the underlying mechanisms, such as the effects on lineage conversion between basal, luminal, BCSCs, mammary stem cells, and the behind cancer driver genes or stemness and fate-determinant factors other than Ki-67 (PMID: 26340526, PMID: 28062556, PMID: 31013830). The effects of phytoestrogens on advanced cancer with heterogeneity are also the concerned issues. 

More importantly, the authors should justify the strength of evidence provided by the literatures instead of just mention the observed results of these studies, even though the authors list more than 200 references. Actually, many results of these studies are controversial because these results are influenced by many confounding factors that should be included for consideration and discussed.

It is suggested to provide authors’ personal opinions for the future direction of investigation to overcome such controversial effect of phytoestrogen on BCSCs.

3.1.4 “These anticancer effects of genistein are attributed to estrogen-independent pathways, such as caspase activation, tyrosine kinase inhibition, and epigenetic modifications.” à explain more, what kinds of epigenetic modifications? What tyrosine kinases are inhibited?

Have a table for in vitro studies to show the differential dose-effects of these phytoestrogens because many of them exhibit both tumor promotion and suppression effects depending on the doses. Please also indicate the concentrations and the cell types when describe their biological activities, such as in section 3.1.7, “Naringenin was found to inhibit the growth and migration of both ER-negative and ER-positive breast cancer cells in vitro and in vivo…”

Figure 1B legend, can be explained in more detail.

Tables are too small to read.

Typo error:

Page 6, section 3.1.5, the neighboring ER-“nehative” BCSCs

Response: The review in our opinion provides a global and unbiased review of the literature on the topic of phytoestrogens and breast cancer stem cells. As this is a short review and not a book on the topic, we feel that it has provided more than enough detail and information for readers to understand and appreciate the topic.

In addition, we specifically provided details on the types of signaling pathways (VEGF, PTK, and MAPK) either activated or repressed by phytoestrogens to inhibit breast cancer stem cells.

We have revised the figure legend to provide more detail regarding the figure. The font of the table is Arial #11 which should be sufficient.

The spelling error has been corrected.

Thank you,
Clodia

Reviewer 2 Report

The manuscript is a review article on phytoestrogens as potential preventive or therapeutic agents against breast cancer, and their ability to affect Breast Cancer Cancer Stem Cells (BCCSC)

The manuscript is well organized, provides the reader with enough background before presenting this class of molecules and both preclinical and clinical results obtained so far

These results are critically discussed, and so is the possible role of phytoestrogens as anti-breast cancer agents

228 references support the manuscript

There are no major revisions requested, this reviewer, however, has two questions:

  • is there any side by side ex vivo comparison of the efficacy against BCCSC of traditional antiestrogens (i.e., tamoxifen, fulvestrant) vs phytoestrogens?
  • can phytoestrogens circumvent resistance of BCCSC to antiestrogens?

There are a few typos

  • 3.1.2 first line “it known” should be “it is known”?
  • 3.1.5 on page 5 “ERb and inhibition of PPRg and FASN expression” the font is different from the rest of the article
  • 3.1.5 on page 6 “ER nehative” should be “ER negative”

Author Response

Dear Review,

Thank you for your valuable comments and questions.

In answer to your questions:

  1. Is there any side by side ex vivo comparison of the efficacy against BCCSC of traditional antiestrogens (i.e., tamoxifen, fulvestrant) vs phytoestrogens?

During our search of the literature, we found several articles with indirect evidence that certain phytoestrogens (i.e. genistein, dadzein, or naringenin) enhance anti-tumor efficacy of tamoxifen in rat models or genetically modified models of drug resistance (Zhang et al., 2017; Tonetti et al., 2007; Reiter et al., 2004, and Limer et al., 2006). To our knowledge, the only article that conducted studies on BCSCs comparing tamoxifen, fulvestrant to phytoestrogens was Peiffer et al., NPJ Breast Cancer 2020. 

2. Can phytoestrogens circumvent resistance of BCCSC to antiestrogens? 

Please see answer to #1.

We corrected all typos and fonts.

Thank you,
Clodia

Round 2

Reviewer 1 Report

This reviewer can accept the rebuttal provided by the authors.